# Clinical Validation of an Artificial Intelligence Model for Detecting Distal Radius, Ulnar Styloid, and Scaphoid Fractures on Conventional Wrist Radiographs

**DOI:** 10.3390/diagnostics13091657

**Published:** 2023-05-08

**Authors:** Kyu-Chong Lee, In Cheul Choi, Chang Ho Kang, Kyung-Sik Ahn, Heewon Yoon, Jae-Joon Lee, Baek Hyun Kim, Euddeum Shim

**Affiliations:** 1Department of Radiology, Korea University Anam Hospital, Seoul 02841, Republic of Korea; 2Department of Orthopedics Surgery, Korea University Anam Hospital, Seoul 02841, Republic of Korea; 3Crescom Inc., Seongnam 13493, Republic of Korea; 4Department of Radiology, Korea University Ansan Hospital, Ansan 15355, Republic of Korea

**Keywords:** artificial intelligence, convolutional neural network, distal radius fracture, ulnar styloid fracture, scaphoid fracture

## Abstract

This study aimed to assess the feasibility and performance of an artificial intelligence (AI) model for detecting three common wrist fractures: distal radius, ulnar styloid process, and scaphoid. The AI model was trained with a dataset of 4432 images containing both fractured and non-fractured wrist images. In total, 593 subjects were included in the clinical test. Two human experts independently diagnosed and labeled the fracture sites using bounding boxes to build the ground truth. Two novice radiologists also performed the same task, both with and without model assistance. The sensitivity, specificity, accuracy, and area under the curve (AUC) were calculated for each wrist location. The AUC for detecting distal radius, ulnar styloid, and scaphoid fractures per wrist were 0.903 (95% C.I. 0.887–0.918), 0.925 (95% C.I. 0.911–0.939), and 0.808 (95% C.I. 0.748–0.967), respectively. When assisted by the AI model, the scaphoid fracture AUC of the two novice radiologists significantly increased from 0.75 (95% C.I. 0.66–0.83) to 0.85 (95% C.I. 0.77–0.93) and from 0.71 (95% C.I. 0.62–0.80) to 0.80 (95% C.I. 0.71–0.88), respectively. Overall, the developed AI model was found to be reliable for detecting wrist fractures, particularly for scaphoid fractures, which are commonly missed.

## 1. Introduction

The wrist is one of the most common sites for fractures. According to previous studies, the annual incidence of hand and wrist fractures is approximately 18 million worldwide [1,2]. Wrist fractures typically indicate the distal radius and ulnar bone, due to their high frequency [3]. Although scaphoid fractures are less frequent than the previous two fractures, they account for the majority (about 82–89%) of carpal bone fractures [4]. Despite their high prevalence, wrist fractures are often missed on radiographs. In particular, scaphoid fractures are difficult to detect, and if a scaphoid fracture is not detected and properly treated, non-union can occur, leading to serious complications such as avascular necrosis and, eventually, to loss of function [5,6,7]. Therefore, to prevent any missed fractures, it is crucial to detect any abnormality during the initial radiographic evaluation at the patient’s first visit.

Radiographic interpretation errors are influenced by human and environmental factors, such as the fatigue or inexperience of a clinician [8]. Since artificial intelligence (AI), unlike human activity, is indefatigable and relatively consistent, interpretation errors may be reduced with the use of AI. Therefore, several deep-learning-based fracture detection models have been developed and several studies have demonstrated the feasibility of fracture detection models on radiographs [9]. One of the deep learning techniques, known as the convolutional neural network (CNN), is widely used because it is capable of “learning” the discriminative features of pixel information on large image sets in order to fit a diagnostic problem. Thian et al. showed that the CNN model can detect and localize radius and ulna fractures on wrist radiographs with high sensitivity and specificity [8]. Hendrix et al. demonstrated that the AI-based scaphoid fracture model can achieve radiologist-level performance in detecting a scaphoid fracture on hand and wrist radiographs [10].

To our knowledge, there is no AI-based solution that can detect all three frequently encountered and clinically important wrist fractures: distal radius, ulnar styloid, and scaphoid fractures. Therefore, we developed an AI-based wrist fracture detection software that is able to detect and mark the fracture sites.

The purpose of this study is to verify the performance of the wrist-fracture detection AI model and to evaluate the clinical feasibility of the model.

## 2. Materials and Methods

This study was reviewed and approved by the institutional review board and the ethics committee of Korea University Anam Hospital on 26 August 2019 (IRB number: 2019AN0385).

This study was based on three steps: model development, clinical testing, and statistical analysis. Details about each step and the study population are provided in Figure 1.

### 2.1. Model Development

As summarized in Figure 2, the model consists of five modules with four analytical steps: (1) data input and preprocessing, (2) detection of automatic regions of interest (ROIs), (3) location segmentation and fracture classification, and (4) integrated assessment results.

The first step was data input and preprocessing. Image preprocessing included CLAHE, normalization, histogram matching, and sharpening. In the second step, a deep learning-based detection model named RetinaNet, which applies focal loss to improve the class imbalance and make backgrounds easy to predict [11], was used to automatically detect each ROI (radius, ulnar styloid, and scaphoid). In the third step, segmentation of the fracture site and fracture classification were performed. We used the DeepLab v3 model for segmentation of the AI module and the NasNet model for the fracture classification module. NasNet is based on automated machine learning (AutoML) that automatically calculates the location and combination of variable operations, such as the convolution layer and the average pooling, to make optimal models [12]. Finally, the results of both the segmentation and fracture classification modules were integrated and our AI model showed a final decision (fracture or no-fracture), including probability. The final decision was determined by comparing the probability value of the segmentation analysis result and the probability value of the classification analysis result with the threshold value of each part. Each of the four steps was fully automatic. The model was implemented using an open-source machine learning library, TensorFlow version 1.9.0, and Keras 2.2.2. Sample images of MediAI-FX, the automatic solution interface of this software, are shown in Figure 3.

We trained the model using the MURA dataset, one of the largest public musculoskeletal radiographic image datasets, which was published by the Stanford ML group in January 2018, together with images from our hospital. A total of 3791 wrist radiographs from MURA were used for distal radius and ulnar styloid fracture training, while 641 radiographs from our hospital were used for scaphoid fracture training. We used 20% of the dataset as a validation set.

### 2.2. Clinical Test

For the clinical test, subjects were selected by simple random sampling from among the patients who visited the emergency department and underwent wrist radiographs to evaluate wrist fractures at the Korea University Anam Hospital between January 2010 and February 2020. Prior to the sampling process, a musculoskeletal radiologist with 19 years of clinical experience (reviewer 1) reviewed all the radiographs and excluded those with anatomic variations or bony abnormalities other than fractures, such as a bone tumor. The reviewer also excluded radiographs with poor imaging quality, inadequate fields of view, or foreign bodies, including orthopedic hardware, splints, or casts. As a result, a total 1186 wrist studies from 593 patients were selected for the clinical test.

In our institution, wrist radiography for patients with trauma is performed bilaterally and consists of anteroposterior (AP) and lateral views. Our emergency department clinicians also commonly take oblique views to aid in carpal bone fracture detection, so we included all these views (AP, lateral, and oblique) in our study.

To establish a ground truth, two human experts independently evaluated the fractures on each wrist radiograph by using a self-developed platform (e-CRF). Reviewer 1 was a musculoskeletal radiologist with 19 years of clinical experience, and reviewer 2 was an orthopedic upper-limb surgeon with 19 years of clinical experience. Both reviewers marked all fracture sites with a bounding box. Nothing was marked if there was no fracture. Each image was analyzed independently; thus, only the view where the fracture was visible was marked. For instance, if a fracture was detected on the AP view but not on the lateral view, it was only marked on the AP view. If there was a significant discrepancy between the two reviewers, a consensus meeting was held to resolve it. In cases where there was still no agreement after the meeting, the decision was made based on follow-up radiography or CT.

For evaluating clinical efficacy, a two-year-fellowship-trained musculoskeletal radiologist (reviewer 3) and a one-year-trained radiology resident (reviewer 4) independently evaluated the wrist fractures in two different sessions. Initially, both reviewers marked fracture sites in the same manner without AI model assistance. Three weeks after the washout period, they repeated the fracture analysis with AI model assistance. The total interpretation time for each session was recorded on the self-developed platform. Finally, the clinical research coordinator conducted the fracture detection using the AI model and reviewer 1 confirmed the model’s results on a heatmap.

### 2.3. Statistical Analyses

To evaluate the model’s performance, we calculated sensitivity, specificity, and accuracy by comparison with the ground truth. Furthermore, we drew receiver operating characteristic (ROC) curves and analyzed the area the under the curve (AUC) of the model. We determined each value per wrist (i.e., per study including AP, lateral, and oblique views). As radiologists or clinicians usually study multiple wrist radiographs at once to detect fractures, we calculated the values per wrist rather than per image. The values were calculated based on each site (distal radius, ulnar styloid, and scaphoid). The model’s performance was considered equivalent when the lower limits of the 95% confidence interval (C.I.) of sensitivity, specificity, accuracy, and AUC of our model was higher than the lower limits of the 95% C.I. of sensitivity, specificity, accuracy, and AUC from previous studies [6,8].

To evaluate the clinical efficacy of the model, we compared the diagnostic accuracy of the two novice radiologists (reviewer 3 and 4) per wrist, with and without model assistance. In addition, sensitivity, specificity, and AUC were compared. Furthermore, we calculated the amount of time spent and performed via a paired t-test to compare the reading times for reviewers 3 and 4, with and without model assistance.

All statistical analyses were conducted using SAS version 9.4 (SAS Institute Inc., Cary, NC, USA). A *p*-value < 0.05 was considered statistically significant.

## 3. Results

### 3.1. Clinical Test Cohort

A total of 593 subjects were enrolled in this study, including 294 males and 299 females. Among them, 398 subjects had at least one site fracture, with 332 having a radius fracture, 270 having an ulnar styloid fracture, and 32 having a scaphoid fracture. The mean age of all subjects was 52.7 ± 19.9 years. Thirty-two of the subjects were under the age of 21, 10 of whom had fractures, including six with radius fractures, five with ulnar styloid fractures, and four with scaphoid fractures.

### 3.2. Model Performance

Table 1 shows the sensitivity, specificity, accuracy, and AUC per wrist. The sensitivity of distal radius and ulnar styloid fractures was 0.972 (95% C.I. 0.956–0.990, and 0.977 (95% C.I. 0.959–995), respectively. The specificity of distal radius and ulnar styloid fractures was 0.832 (95% C.I. 0.807–0.957) and 0.873 (95% C.I. 0.851–0.894), respectively. The AUC of distal radius and ulnar styloid processes was 0.903 (95% C.I. 0.887–0.918) and 0.925 (95% C.I. 0.911–0.939), respectively. All values were higher than the equivalent values. The sensitivity and specificity of scaphoid fractures were 0.870 (95% C.I. 0.760–0.989), and 0.740 (95% C.I. 0.714–0.765), respectively. Table 2 shows the variable sensitivity and specificity of scaphoid fracture detection according to five previous studies [13,14,15,16,17,18]. The AUC of scaphoid fracture of our model was 0.808 (95% C.I. 0.748–0.967).

### 3.3. AI Model Clinical Efficacy

Table 3 shows the sensitivity, specificity, accuracy, and AUC determined by two novice radiologists with or without software assistance. The wrist-fracture detection sensitivity determined by reviewers 3 and 4 increased with model assistance in all sessions. In particular, the scaphoid fracture sensitivity determined by the two novice radiologists significantly increased from 0.50 (95% C.I. 0.32–0.68) to 0.72 (95% C.I. 0.53–0.86) and from 0.47 (95% C.I. 0.29–0.65) to 0.66 (95% C.I. 0.47–0.81), respectively. The scaphoid fracture AUC determined by the ROC analysis of the two novice radiologists also significantly increased from 0.75 (95% C.I. 0.66–0.83) to 0.85 (95% C.I. 0.77–0.93) and from 0.71 (95% C.I. 0.62–0.80) to 0.80 (95% C.I. 0.71–0.88), respectively. The scaphoid fracture specificity determined by reviewer 4 slightly decreased; therefore, the scaphoid fracture detection accuracy of reviewer 4 slightly decreased from 0.94 (95% C.I. 0.93–96) to 0.93 (95% C.I. 0.92–0.95). However, this difference was not statistically significant.

The mean interpretation time (seconds), with and without model assistance, was 4.46 ± 6.96 and 5.10 ± 8.37, respectively. Statistically, it was significantly different (*p* < 0.007). The time took approximately 1.1 times longer with model assistance than without model assistance.

## 4. Discussion

Our study demonstrated the sensitivity, specificity, accuracy of the AI-based distal radius, ulnar styloid process, and scaphoid fracture-detection model. Thian et al. suggested an AI-based model for detecting and localizing radius and ulna fractures on wrist radiographs, including frontal and lateral views. Their model’s sensitivity, specificity, and AUC were 0.981 (95% C.I. = 0.956–0.994), 0.729 (95% C.I. = 0.671–0.782), and 0.895 (95% C.I. = 0.876–0.94), respectively. However, they did not distinguish between distal radius and ulna fractures [8]. Carpenter et al. summarized the scaphoid fracture detection accuracy of previous studies (Table 2) [6], and we used those values as equivalent values. According to our study, for each fracture site, the lower 95% C.I. of the AUC of distal radius and ulnar styloid process (0.887 and 0.911, respectively) was higher than the equivalent value of 0.876. In the case of scaphoid fracture detection, the lower value of 95% C.I. of scaphoid fracture detection sensitivity was 0.760, which was higher than the equivalent value of 0.710, which was the second lowest value in previous studies, as shown in Table 2 [15]. In addition, the lower value of 95% C.I. of scaphoid fracture detection specificity was 0.714, which was higher than the equivalent value of 0.360, which was the second lowest value in previous studies, as shown in Table 2 [13]. Furthermore, these values were the same as or higher than those in previous studies, except for the study by Cetti et al. [17]. Thus, our model’s performance for detecting wrist fractures was as accurate as the detection in previous studies.

Moreover, our results proved that the AUC for wrist fracture detection by novice radiologists improved, except for reviewer 3’s detection of ulnar styloid fractures, and it improved more significantly with respect to scaphoid fracture detection. The accuracy was equal to or improved with AI model assistance, except for reviewer 4’s scaphoid fracture detection. This was because reviewer 4’s sensitivity for scaphoid fracture detection improved, but the specificity decreased. However, the decrease in specificity was not statistically significant, whereas the improvement in sensitivity for scaphoid detection was statistically significant. Therefore, we expect that our AI model could be helpful for fracture detection screening in an emergency room setting, where immediate radiographic interpretation by a musculoskeletal radiologist or a wrist surgeon is not available.

Although the model’s assistance improved the diagnostic performance of novice radiologists, the reading time was lengthened by approximately 110%. While this increase was statistically significant (*p* = 0.007), the actual added time was less than 1 s. In contrast to our findings, Hendrix et al. focused on AI-assisted scaphoid fracture detection and found that it reduced the reading time for four out of five experienced musculoskeletal radiologists [18]. However, their study involved skilled radiologists with 5 to 26 years of experience and focused only on scaphoid fracture. Our study may have resulted in slightly longer reading times because we considered two additional areas beyond scaphoid fractures. Furthermore, it is expected that the use of improved user interfaces and better interworking of PACS systems will help to reduce the interpretation time utilizing the AI model. Consequently, our study highlights the need for the improvement of workflow in the development of an assistant AI model.

Recently, AI tools have received significant attention in musculoskeletal radiology, and fracture detection is one of these fields [9,19]. Olczak et al. trained an AI model with approximately 256,000 wrist, hand, and ankle radiographs, and its fracture classification accuracy was 83% [20]. However, that study included ankle radiographs, and the diverse training set may have lowered the accuracy. Recently, there have been more studies that focus on wrist-fracture detection [8,10,18,21,22,23]. In our study, we achieved per-wrist accuracies of 87.2% and 89.6% for distal radius and ulnar styloid process fractures, respectively. These values are comparable to the overall accuracy of distal radius and ulna fracture detection in a previous study, which were 88.9% for frontal X-ray and 91.2% for lateral X-ray [8]. The AUCs in our study for distal radius and ulnar styloid process fracture detection were 0.903 and 0.925, respectively. The AUC value of Kim and MacKinnon’s CNN model, which showed binary results (the presence or absence of a fracture) and did not show localization, was 0.956 [22]. Although our model’s AUC was slightly lower, our model marked the fracture site with heatmap images, which could particularly resolve the “black-box” nature of CNNs [24]. We retrospectively reviewed all heatmap images of our model’s results and there were no serious errors. Therefore, our model’s heatmap images could assist clinicians in determining whether a marked fracture is true or false. Langerhuizen et al. demonstrated a scaphoid fracture-detection model with 72% accuracy, 84% sensitivity, and 60% specificity, which were similar to the sensitivity and accuracy of orthopedic surgeons, but lower than their specificity [21]. Our model had slightly higher overall accuracy.

One of the strengths of our model is its ability to analyze all three high-frequency fractures simultaneously, making it the first such model, to our knowledge. A recent study developed an AI model for the detection of all wrist fractures; however, the performance of that model in detecting carpal bones other than scaphoid, such as pisiform and trapezium, was poor [23]. Hence, we believe that our model is helpful in accurately detecting wrist fractures relatively more frequently. Moreover, we incorporated the oblique view of wrist radiographs, an important view for detecting wrist fractures. Consequently, we expect our model to be beneficial in a clinical setting.

Our study had some limitations. First, it was a single-center retrospective study with mostly adult subjects (94.6%). Wrist fractures can be found in children, particularly distal radius fractures that often occur in children and adolescents under the age of sixteen [25], and the fracture pattern of children may differ from that of adults [26]. Therefore, multicenter prospective studies involving children and adolescents are needed. Second, since our study was unable to detect fractures other than distal radius, ulnar styloid, and scaphoid fractures, such as other carpal fractures, trauma-related carpal alignment abnormalities, or even bone tumors or infections, interpretation by a radiologist is necessary. However, as a fracture screening tool, our model is adequate, and it is not intended to be used independently in a clinical setting. Third, we did not distinguish between recent and old fractures in our study. In our training, old ulnar styloid and scaphoid fractures in a non-union state were classified as fracture-positive, whereas old radius fractures a healing deformity were classified as fracture-negative. In a clinical setting, it is important to differentiate between recent and old fractures. Hence, further studies are required.

Finally, the ultimate validation of the clinical feasibility of our AI model requires a demonstration of clinical outcomes beyond the performance metrics, such as reducing complications or lowering medical costs. In the future, it will be necessary to determine whether our AI model has improved patients’ outcomes or medical cost efficacy in real emergency room situations.

## 5. Conclusions

The AI model we developed for detecting simultaneously three commonly encountered and clinically important wrist fractures (distal radius, ulnar styloid, and scaphoid fractures) was reliable for fracture detection with high accuracy. It was particularly useful for detecting the easily overlooked scaphoid fractures, especially by inexperienced radiologists. We expect that the model will be useful and time-saving in the future, once it is fully integrated with the PACS system.

## Figures and Tables

**Figure 1 diagnostics-13-01657-f001:**
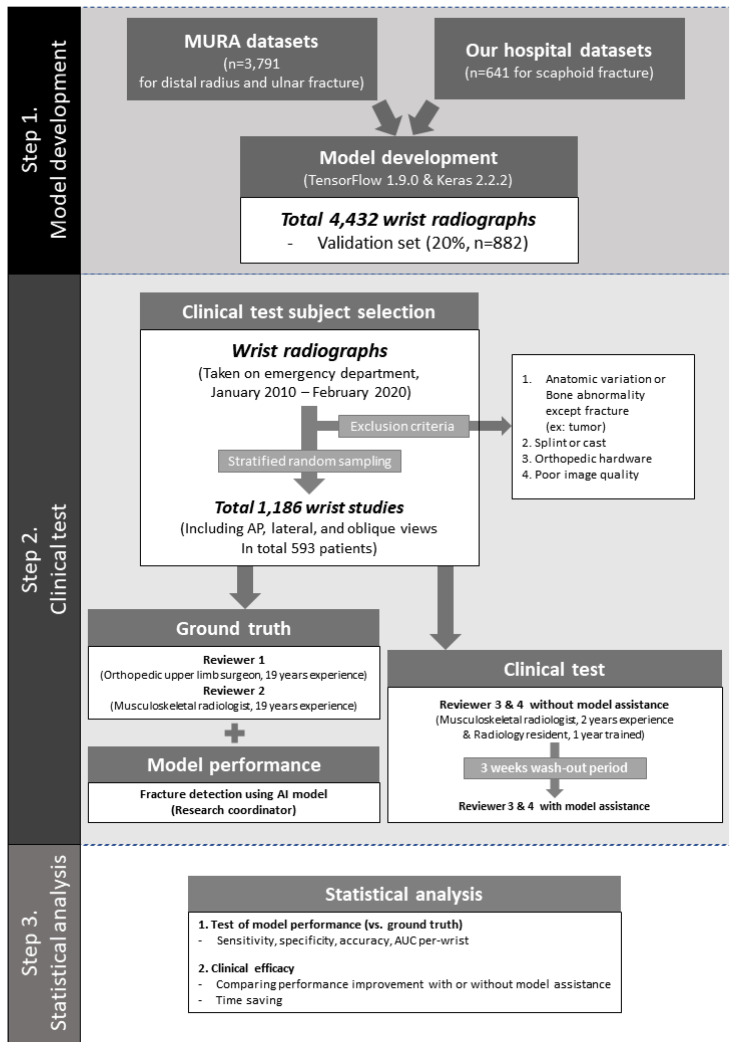
Overview of the study. The model was developed using two datasets. A total of 4432 wrist radiographs were used as training and validation sets, and a total of 1186 wrist radiographs were used for clinical testing. A statistical analysis was conducted.

**Figure 2 diagnostics-13-01657-f002:**
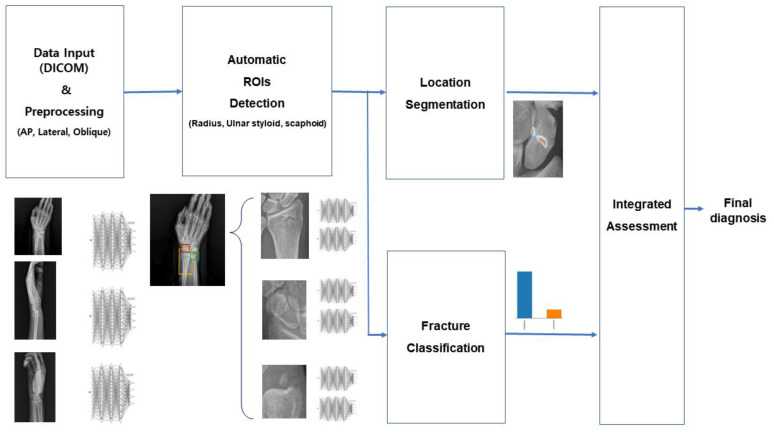
Overview of wrist-fracture (radius, ulnar styloid, and scaphoid) detection models. DICOM = digital imaging and communications in medicine. ROI = region of interest.

**Figure 3 diagnostics-13-01657-f003:**
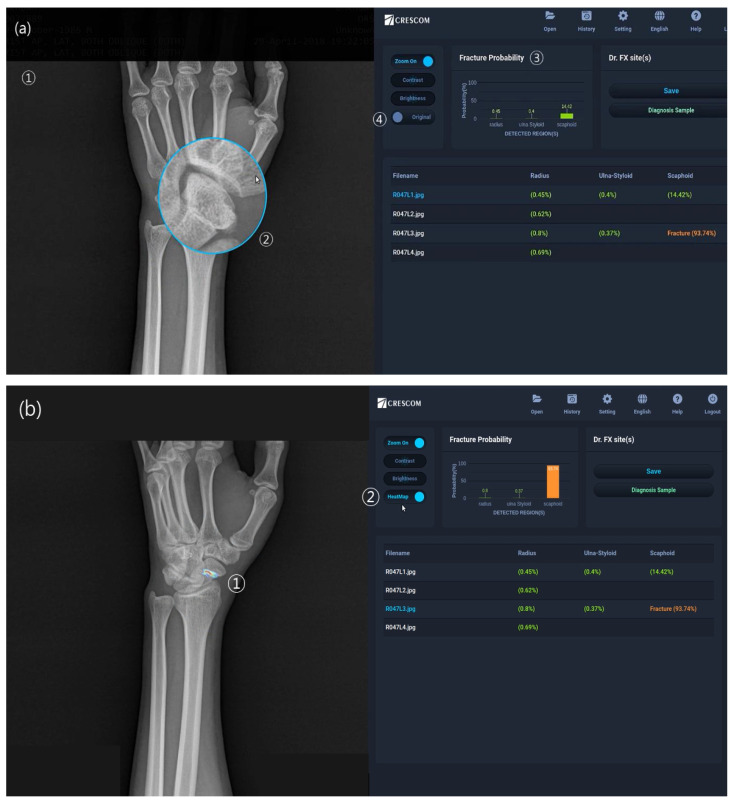
Image of the proposed wrist fracture-detection artificial intelligence model. (**a**) The initial wrist-detection result image, which includes the following: ① wrist-fracture radiography for analysis, ② the magnified image, as indicated by the mouse cursor, ③ the graph showing the fracture probability for radius, ulna, and scaphoid, and ④ a button to display the heatmap in (**b**). (**b**) The heatmap image, which includes ① a heatmap indicating the fracture site and ② a button to return to the original image in (**a**).

**Table 1 diagnostics-13-01657-t001:** The sensitivity, specificity, accuracy, and AUC per wrist.

	Sensitivity (95% C.I.)	Specificity (95% C.I.)	Accuracy (95% C.I.)	AUC (95% C.I.)
Distal radius	0.972 (0.956–0.990)	0.832 (0.807–0.957)	0.872 (0.852–0.890)	0.903 (0.887–0.918)
Ulnar styloid	0.977 (0.959–0.995)	0.873 (0.851–0.894)	0.896 (0.878–0.913)	0.925 (0.911–0.939)
Scaphoid	0.870 (0.760–0.989)	0.740 (0.714–0.765)	0.740 (0.715–0.765)	0.808 (0.748–0.967)

Note: C.I. = confidence interval, AUC = area under the curve.

**Table 2 diagnostics-13-01657-t002:** The sensitivity and specificity of scaphoid fracture based on X-ray.

	Sensitivity (95% C.I.)	Specificity (95% C.I.)
Cetti 1982 [17]	0.94 (0.86–0.98)	0.90 (0.85–0.94)
Dias 1987 [16]	0.83 (0.73–0.91)	0.44 (0.30–0.59)
Langer 1988 [15]	0.88 (0.71–0.97)	0.67 (0.61–0.73)
Banerjee 1999 [14]	0.90 (0.76–0.97)	0.78 (0.62–0.89)
Annamalai 2003 [13]	0.50 (0.36–0.64)	0.50 (0.36–0.64)

Note: C.I. = confidence interval.

**Table 3 diagnostics-13-01657-t003:** The sensitivity, specificity, accuracy, and AUC as determined by each reviewer with or without software.

	Reviewer 3		Reviewer 4	
	Without Software	With Software	Without Software	With Software
**Sensitivity (95% C.I.)**				
Distal radius	0.94 (0.91–0.96)	0.97 (0.95–0.99)	0.90 (0.86–0.93)	0.95 (0.92–0.97)
Ulnar styloid	0.97 (0.94–0.98)	0.98 (0.95–0.99)	0.71 (0.65–0.76)	0.86 (0.81–0.90)
Scaphoid	0.50 (0.32–0.68)	0.72 (0.53–0.86)	0.47 (0.29–0.65)	0.66 (0.47–0.81)
**Specificity (95% C.I.)**				
Distal radius	0.97 (0.96–0.98)	0.97 (0.96–0.98)	0.95 (0.94–0.97)	0.94 (0.92–0.96)
Ulnar styloid	0.98 (0.96–0.98)	0.97 (0.95–0.98)	0.99 (0.98–1.00)	0.97 (0.95–0.98)
Scaphoid	0.99 (0.99–1.00)	0.98 (0.97–0.99)	0.96 (0.94–0.97)	0.94 (0.92–0.95)
**Accuracy (95% C.I.)**				
Distal radius	0.96 (0.95–0.97)	0.97 (0.96–0.98)	0.94 (0.92–0.95)	0.94 (0.93–0.95)
Ulnar styloid	0.97 (0.96–0.98)	0.97 (0.96–0.98)	0.93 (0.91–0.94)	0.94 (0.93–0.96)
Scaphoid	0.98 (0.97–0.99)	0.98 (0.97–0.98)	0.94 (0.93–0.96)	0.93 (0.92–0.95)
**AUC (95% C.I.)**				
Distal radius	0.96 (0.94–0.97)	0.97 (0.96–0.98)	0.93 (0.91–0.95)	0.94 (0.93–0.96)
Ulnar styloid	0.97 (0.96–0.98)	0.97 (0.96–0.98)	0.85 (0.82–0.88)	0.91 (0.89–0.93)
Scaphoid	0.75 (0.66–0.83)	0.85 (0.77–0.93)	0.71 (0.62–0.80)	0.80 (0.71–0.88)

Note: C.I. = confidence interval, AUC = area under the curve.

## Data Availability

The MURA dataset is available at https://stanfordmlgroup.github.io/competitions/mura/ (accessed on 7 May 2023).

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
