# Peer review of "Clinical Validation of an Artificial Intelligence Model for Detecting Distal Radius, Ulnar Styloid, and Scaphoid Fractures on Conventional Wrist Radiographs"

_diagnostics, 2023, doi:10.3390/diagnostics13091657_

Round 1
Reviewer 1 Report
Many thanks to the authors for revising the manuscript.
This piece is relavant and interesting for readers.
Author Response
Thank you for reviewing our paper and also for the great comments! Thanks to you, we were able to achieve good results.
Reviewer 2 Report
- In the title, it is written "Scaphoid Factures". But it should be Fractures.
- References used in this manuscript are from the year 2021 and before while there are similar works published in 2022 and 2023. Authors could discuss the latest similar works and conduct a comparative analysis to highlight the pros and cons of the proposed model with the latest existing solutions.
Author Response
- In the title, it is written "Scaphoid Factures". But it should be Fractures.
--> Thank you for your comment. We changed our title as follows: Clinical Validation of AI-based Model for Detecting Distal Radius, Ulnar Styloid, and Scaphoid Fractures on Conventional Wrist Radiographs
- References used in this manuscript are from the year 2021 and before while there are similar works published in 2022 and 2023. Authors could discuss the latest similar works and conduct a comparative analysis to highlight the pros and cons of the proposed model with the latest existing solutions.
--> Thank you for your great comment. We found two recent relevant papers (Reference 18 & 23) and added them in our discussion as follows;
- In contrast to our findings, Hendrix et al. focused on AI-assisted scaphoid fracture de-tection and found that it reduced reading time for 4 out of 5 experienced musculoskel-etal radiologists (18). However, their study involved skilled radiologists with 5 to 26 years of experience and focused only on scaphoid fracture. Our study may have re-sulted in slightly longer reading times because we considered two additional areas be-yond scaphoid fractures.
- A recent study developed an AI model for the detection of all wrist fractures, however, the performance of model in detecting other carpal bones except scaphoid, such as pis-iform, trapezium was poor (23). Hence, we believe that our model is helpful in accu-rately detecting relatively more frequent wrist fractures.
Reviewer 3 Report
Thank you for allowing me to peer review.
The developed AI model was found to be reliable for detecting wrist fractures, particularly for scaphoid fractures, which are commonly missed.
I think this paper is fine with minor revision.
Please show the Xp for each of the distal radius, ulnar hyoid process, and navicular fractures that were rated differently by the examiner and AI.
Are orthopedic surgery residents not included in the examiner?
Quality of English Language is no problem.
Author Response
- Please show the Xp for each of the distal radius, ulnar hyoid process, and navicular fractures that were rated differently by the examiner and AI.
--> Thank you for your comment. There were few cases that read differently between AI and the examiners. In cases of ulnar styloid fracture, AI may not always detect it on lateral views, despite being well-detected on AP or oblique views. Additionally, there are few instances where AI considers non-union as a fracture. Compared to AI, examiners are relatively better at distinguishing these factors. In cases of scaphoid fracture, AI often fails to detect the fracture on lateral and oblique views. This is thought to be due to the training process primarily focusing on AP views.
- Are orthopedic surgery residents not included in the examiner?
--> Thank you for your comment. Unfortunately, orthopedic surgery residents were not included in our study. For orthopedic surgery residents, it is believed that this program would be more helpful, as they can actually examine the patient and correlate the X-rays with the pain site. However, further research on patient outcomes in actual clinical setting is needed, and this has been noted as a limitation in our paper.
Reviewer 4 Report
Dear authors i am happy to review your excellent work, there are many points to be mentioned in manuscript to make it more clear to the readers. Please see the comments attached in file.

Author Response
We would like to thank the editor and reviewers for their time and effort devoted to reviewing and commenting on this manuscript. We have made revisions and clarifications according to the reviewers’ suggestions which have helped us improve the quality of our manuscript.
We replied to all comments and revised word file.

Round 2
Reviewer 4 Report
Dear authors excellent job. Well done.